# Do Nutrients and Nutraceuticals Play a Role in Diabetic Retinopathy? A Systematic Review

**DOI:** 10.3390/nu14204430

**Published:** 2022-10-21

**Authors:** Agostino Milluzzo, Martina Barchitta, Andrea Maugeri, Roberta Magnano San Lio, Giuliana Favara, Maria Grazia Mazzone, Laura Sciacca, Antonella Agodi

**Affiliations:** 1Department of Clinical and Experimental Medicine, Endocrinology Section, University of Catania, 95122 Catania, Italy; 2Department of Medical and Surgical Sciences and Advanced Technologies “GF Ingrassia”, University of Catania, Via S. Sofia 87, 95123 Catania, Italy; 3Research, Preclinical Development and Patents, SIFI S.p.A., Lavinaio-Aci S. Antonio, 95025 Catania, Italy

**Keywords:** retinopathy, diabetes mellitus, diabetes complications, nutraceuticals, vitamins, supplements, antioxidants, microvascular complications, visual acuity, blindness

## Abstract

Diabetic retinopathy (DR) is a multifactorial neuro-microvascular disease, whose prevalence ranges from 25% to 60% of subjects affected by diabetes mellitus, representing the main cause of legal blindness in adults of industrialized countries. The treatment of advanced stage of DR is based on invasive and expensive therapies, while few strategies are available for the early stage or prevention. The mechanisms underlying DR involve a complex interplay between the detrimental effects of hyperglycemia, dyslipidemia, hypoxia, and oxidative stress, providing several pathways potentially targeted by nutrients and nutraceuticals. In this study, we conducted a systematic review of observational and interventional studies, evaluating the effect of nutrients and/or nutraceuticals on the risk of DR and their potential use for the treatment of patients with DR. The analysis of the 41 included studies (27 observational and 14 interventional studies) suggests a promising preventive role of some nutrients, in particular for vitamins B (i.e., B1 and B12), D, and E. However, further investigations are necessary to clarify the potential clinical application of nutraceuticals in the prevention and treatment of DR.

## 1. Introduction

Diabetic retinopathy (DR) is a progressive neuro-microvascular complication of both type 1 (T1D) and type 2 diabetes mellitus (T2D) [1]. While the early damage of retinal small blood vessels and neurons could be asymptomatic, the advanced stage of DR leads to a vision loss—representing the main cause of legal blindness in industrialized countries among the working age adults—and to a reduced daily functionality and quality of life [2,3]. With the increasing prevalence of diabetes mellitus (DM), the worldwide number of people affected by DR is estimated to reach 200 million in the next decade, thus becoming a main public health and economic burden [4]. The prevalence of DR in observational studies ranges from 25% to 60% according to the country and the type of diabetes, appearing higher in T1D rather than T2D [5,6,7,8]. Both the improvement of community screening programs and the awareness of DR risk factors are mandatory to reduce the incidence of DR and related visual impairment [9]. Genetic, epigenetic and environmental factors contribute to the etiopathogenesis of DR [10,11,12]. The main determinants of DR, most of them in common with diabetes nephropathy, neuropathy and macrovascular complications, are related to the course of both diabetes—diseases duration and glucose control—and comorbidities, such as hypertension, dyslipidaemia, renal impairment, and overweight/obesity [13,14,15,16,17,18,19]. Nevertheless, some diabetic subjects without these risk factors can likewise develop this complication, suggesting the role of other pathogenetic elements [17,20]. Among modifiable risk factors, unhealthy lifestyles, such as dietary intake, physical activity, tobacco consumption could influence the course of DR [21,22]. In particular, preclinical and clinical studies indicated a possible role of vitamins and micro- or macro-nutrients intake—through usual feeding or dietary supplementation—in the course of DR and its therapy [23,24]. However, current evidence is heterogeneous and controversial, raising the need for a deep evaluation of the role of nutrients and nutraceuticals in DR. For this reason, we carried out a systematic review of both observational and interventional studies, which evaluated the effect of nutrients and/or nutraceuticals on the risk of DR and their potential use for the treatment of patients with DR.

## 2. Materials and Methods

The present systematic review was carried out in accordance with the Preferred Reporting Items for Systematic Reviews and Meta-analyses (PRISMA) statements and the Cochrane Handbook’s guidelines [25,26] (the PRISMA Checklist is reported in the Appendix A).

### 2.1. Literature Search

Clinical studies evaluating the association between DR and vitamins and/or nutrients were systematically searched on PubMed-Medline (PubMed.gov: available online: https://pubmed.ncbi.nlm.nih.gov/, accessed on 24 November 2021), and Web of Science databases (https://www.webofscience.com/wos/woscc/basic-search, accessed on 24 November 2021), from inception to November 2021. The search strategy applied the following combination of terms: (diabetes retinopathy OR diabetic retinopathy) AND (neuroprotection OR neur* OR nerv*) AND (vitamin* OR supplement* OR nutraceutical* OR antioxidant*). Reference lists of potentially eligible articles were also screened.

### 2.2. Selection Criteria

The following inclusion criteria must have been satisfied: (1) observational or interventional studies; (2) evaluating the role of nutrients or supplements/nutraceuticals (3) in the prevention and/or treatment of DR, (4) among patients with T1D or T2D. Accordingly, the main outcomes of interest were: the risk of developing DR, also considering its clinical forms (i.e., Proliferative Diabetic Retinopathy, PDR; Non Proliferative Diabetic Retinopathy, NPDR); DR stages, severity, and progression; clinical parameters related to DR (e.g., retinal nerve fibre layer thickness, central corneal nerve fibre length, retinal blood flow, best corrected visual acuity, etc.). Only studies published in peer-reviewed journals in English language were included, with no limitations with regard to publication date. Conversely, we excluded previous systematic reviews and meta-analyses, commentary articles, and editorials.

### 2.3. Data Extraction and Quality Assessment

Two authors, applying the selection criteria described above, independently reviewed titles and abstracts of all identified articles. The full texts of all eligible articles were further reviewed to assess whether selection criteria were fully met. Eventual controversies were solved consulting a third author. Two investigators extracted the following information: first author, study year, location, study design, sample size, type of DM, type, dosage, and administration schedule of vitamins or nutraceuticals, methods to explore retinal status, main findings. The quality assessment of the included studies was performed by using the Newcastle-Ottava Scale (NOS). Each study was judged on eight items, categorized into three groups: the selection of the study groups (maximum 4 points); the comparability of the groups (maximum 2 points); and the ascertainment of either the exposure or outcome (maximum 3 points).

## 3. Results

### 3.1. Characteristics of the Included Studies

A total of 840 articles were screened once duplicates had been removed. According to selection criteria, 800 articles were excluded after reading article title and abstract: of these, 286 were not conducted on humans, 269 were literature reviews or meta-analyses, and 245 did not focus on vitamins/nutraceuticals (*n* = 159) or DR (*n* = 86) (Figure 1). In addition, 13 articles were identified from references. Out of the 53 articles that underwent the full-text screening, 12 were excluded for the reasons detailed in Figure 1; therefore, 41 studies investigating the relationship between vitamins/nutraceuticals and DR were included in the systematic review.

Most of the included studies were observational (*n* = 27) and evaluated the amount of specific vitamins in biological samples (blood, urine, saliva, aqueous humour, exhaled breath condensate) or estimated their dietary intake using a food frequency questionnaire (Table 1). The Appendix A provides more details on demographic and clinical characteristics of subjects, according to the diagnosis and degree of DR. Instead, 14 studies—8 of them randomized and controlled (RCT)—evaluated the effect of vitamins/nutraceuticals administration on DR course (Table 2). Study quality was generally high, with 25 studies showing a value of quality assessment of 6 or higher.

Regarding the type of diabetes, the majority of the studies enrolled subjects with T2D (*n* = 28). Among them, one study also included patients affected by impaired fasting glucose (IFG) or impaired glucose tolerance (IGT). Moreover, ten studies evaluated the effect of vitamins/nutraceuticals in T1D retinopathy. In the remaining studies, the type of diabetes is not indicated.

### 3.2. Interventional Studies

Among the 14 intervention studies analysed in this systematic review, 8 were RCT, which evaluated the impact of single vitamins or complex, compared to placebo, on different retinal outcomes in patients with or without DR at baseline (Table 2).

The effect on retinal morphology and function of eye drops containing vitamin B12 and citicoline was explored in 20 patients affected by T1D and mild non-proliferative diabetic retinopathy (NPDR), randomly assigned to receive the treatment or placebo during 3 years of follow-up [27,28]. The placebo group showed a significant increase in inner nuclear layer thickness and a decrease in other plexiform layer thickness and foveal vessel density, while no such worsening was observed in patients treated with vitamin B12 and citicoline, indicating a possible protective role of these molecules [28]. Moreover, while the treated group obtained an improvement of the macular bioelectrical responses detected by multifocal electroretinogram recordings, the placebo group showed a worsening of these parameters [27]. The effects on stabilization or decreased rate of retinal functional impairment, neuroretinal degeneration and microvascular damage could be explained by the citicoline promotion of phospholipids precursors biosynthesis, reactive oxygen species (ROS) scavenging, mitochondrial functionality, and protection from apoptosis through phospholipases A2 inhibition, cardiolipin and sphingomyelin stimulation [27,28].

The effect of Ocufolin, a vitamin complex composed of L-methylfolate (vitamin B9), vitamin B1, B2, B6, B12, C, D, E, N-acetylcysteine and other compounds (Table 2), on DR was tested by Liu and colleagues in a specific cohort of diabetes subjects with methylenetetrahydrofolate reductase (MTHFR) polymorphisms [29,30]. MTHFR, an enzyme involved in the folate methylation, contributes to convert homocysteine to methionine. Its polymorphisms and impaired function cause increased plasma homocysteine levels, hypertension, and increased microvascular retinal damage [30]. In the prospective interventional cohort study, Liu and colleagues observed that after six months of Ocufolin administration, patients with MTHFR polymorphisms (C677T, or A1298C) also affected by NPDR improved their visual acuity, superficial retinal and conjunctival vessel density [29,30]. The authors suggest that the observed improvement of retinal status could be obtained by means of the reduction in homocysteine level, insulin resistance, inflammation and mitochondrial oxidative stress related to the action of the different compounds of the vitamin complex [30]. Smolek and colleagues tested a vitamin B complex composed by vitamin B6, B9, and B12 in ten T2D patients with NPDR. After six months of treatment, an improvement of both retinal oedema and light sensitivity was detected [31]. These results, although observed in a small cohort, confirm the possible protective effect of vitamin B on DR progression. Similarly, the administration of vitamin B6 alone, at a dosage of 50–200 mg daily, was related to a long-term lower incidence of DR [32].

To explore the role of vitamin E on DR, Bursell and colleagues designed an 8-month randomized placebo-controlled crossover trial involving 36 T1D without DR and 9 healthy subjects randomly assigned to either 1800 IU vitamin E per day or placebo for 4 months, followed, after treatment crossover, for a further 4 months [18,33]. After vitamin E treatment, the retinal blood flow significantly increased in diabetic patients and was similar to that of healthy subjects, suggesting that high dose oral vitamin E could be effective in normalizing retinal hemodynamic abnormalities in T1D patients with short disease duration [33]. Inflammation and oxidative stress contribute to the pathogenesis of DR also in T2D, due to the increased inflammatory pattern related to overweight/obesity and metabolic syndrome, typical of these patients. Thus, the antioxidant properties of vitamin E might have a protective role in DR progression, in particular reducing the level of serum malondialdehyde (MDA) as observed in 282 T2D patients with DR who received 300 mg of vitamin E per day for three months [34]. In support of the antioxidant role of vitamin E, a RCT by Domanico and colleagues demonstrated a significant reduction in free oxygen radicals and central macular thickness in patients with NPDR treated, for six months, with vitamin E, pycnogenol and coenzyme Q10 compared to placebo [35].

The Diabetes Visual Function Supplement Study (DiVFuSS), RCT involving 67 patients with both T1D and T2D, evaluated the effect of a vitamin complex composed by vitamin B1, C, D3, E, omega 3 polyunsaturated fatty acid (PUFA), curcumin, zinc, and other compounds (Table 2). After six months, patients on active supplement compared with placebo had a significantly better visual function [36]. The authors indicated the reduction in oxidative stress, inflammation, and mitochondrial damage as possible mechanism underlying the observed visual function improvement.

Some of the elements included in the complex tested in the DiVFuSS study were individually tested in other trials [37,38,39]. Britten-Jones and colleagues randomized 43 T1D patients to receive oral PUFA at a daily dosage of 1800 mg or placebo. After six months, the treated group had a significant improvement of the central corneal nerve fibre length, compared to placebo group, indicating a possible protective role of PUFA in peripheral nerve health [37]. This effect could be mediated by the improvement of lipid panel, inflammation, and neuro-apoptosis [37]. Kheirouri and colleagues investigated whether zinc supplementation could influence the pathogenic mechanisms of DR. Their RCT recruited 50 patients with DR, allocated to receive for three months 30 mg of zinc per day or placebo, to assess the serum level of vascular endothelial growth factor (VEGF), brain-derived neurotrophic factor (BDNF), and nerve growth factor (NGF). The level of the analysed growth factors did not change by the Zn supplementation. Nevertheless, the level of VEGF was negatively related to zinc levels [38].

The anti-inflammatory action of curcumin, homotaurine, and vitamin D3 was tested by measuring the levels of pro-inflammatory cytokines, in particular tumour necrosis factor alpha (TNF-α), interleukin 2 (IL2), interleukin 6 (IL6), and platelet derived growth factor (PDGF) in vitreous samples of diabetic patients with PDR. Although curcumin, homotaurine, and vitamin D3 individually determined a slightly appreciable improvement of the inflammatory panel, when used in combination, this effect was enhanced [39].

An anti-inflammatory role was referred also to crocin, the main component of saffron, tested in a double-masked, placebo controlled, RCT involving 60 patients with diabetic maculopathy and refractory to conventional therapies (macular photocoagulation, intravitreal injection of anti-VEGF agents). Patients in the high-dose crocin group (15 mg per day) significantly improved their visual acuity compared to the placebo, suggesting an antioxidant and neuroprotective role of crocin in diabetic maculopathy and mediated by an increase in blood flow, oxygen supply, mitochondrial genesis and an improvement of inflammatory damage and oxidative [40].

### 3.3. Observational Studies

A total of 10 out of 27 observational studies investigated the influence of vitamin D or its metabolites on the onset or progression of DR (Table 1). Although conflicting results are available, in eight of these studies, an inverse relation between the serum/plasma level of vitamin D and the risk of DR was detected [41,42,43,44,45,46,47,48]. The largest study, involving more than 9000 T2D patients, observed a significantly (*p* = 0.008) higher incidence of microvascular diabetes-related complications, including DR, in patients with baseline 25-hydroxyvitamin D (25OH-D) lower than 50 nmol/L [45]. Noteworthy, this risk excess lost significance at multivariate analysis after adjustment for glycated haemoglobin (HbA1c) and seasonality. Similarly, many others observational studies highlighted a relation between lower level of vitamin D and the prevalence of DR in T2D [42,43,44,46,47,49]. In most of these studies, classification of DR was performed by exploring the retina with ophthalmoscopy [42,43,46,47,49], whereas Gungor and colleagues evaluated the retinal nerve fibre layer, by optical coherence tomography (OCT), observing its thinning in patients with vitamin D deficiency [44]. Our research found a single article concerning the role of vitamin D in T1D [48]. In this cross-sectional study of 517 young patients with T1D and mean age 15 years, the vitamin D deficiency, defined as 25OH-D lower than 50 nmol/L, was associated with a significant, doubled risk of DR. Otherwise, a recent observational study by Zhao and colleagues, performed on 815 T2D Chinese patients, did not show any association between the level of 25OH-D and DR [50].

Recently, a cross-sectional study involving 460 T2D patients focused on vitamin D metabolite ratio (VMR) [41]. In particular, the authors measured the plasma concentration of 25-hydroxyvitamin D3(25(OH)D3) and its metabolites 1,25-dihydroxyvitamin D3(1,25(OH)2D3) and 24,25-dihydroxyvitamin D3(24,25(OH)2D3) and its epimer, 3-epi-25-hydroxyvitaminD3(3-epi-25(OH)D3). Moreover, VMR-1 was calculated as a ratio of 24,25(OH)2D3:25(OH)D3; VMR-2 as a ratio of 1,25(OH)2D3:25(OH)D3; VMR-3 was calculated as a ratio of 3-epi-25(OH)D3:25(OH)D. Interestingly, VMR-1 and VMR-2 were associated with DR [41].

Vitamins C and E may influence the course of DR by affecting some of its pathogenic factors: protein glycosylation, oxidative stress, and retinal blood flow [51]. However, the two observational studies conducted by Millen and colleagues on T2D patients did not observe significant associations between serum levels of these antioxidant vitamins and DR [51,52]. Specifically, in their study published in 2004, the authors evaluated the association of serum vitamin E and C levels with prevalent DR in 998 T2D patients, 20% affected by DR [52]. The same authors obtained similar neutral results on DR assessing the vitamins intake with a food-frequency questionnaire in a cohort of 1353 T2D patients [51]. Conversely, She and colleagues, demonstrated, in a cross-sectional study involving 455 T2D patients who did not supplement any vitamins or nutraceuticals, a slight reduction in DR risk related to a higher dietary intake of both vitamin E (OR 0.97, *p* = 0.036) and selenium (OR 0.98, *p* = 0.017) [53]. To support the possible role of selenium in DR, Zhou and colleagues observed a significantly lower urinary selenium level in T2D patients with DR [54]. In a small cohort 36 subjects, including 24 T2D patients, the authors evaluated the role of oxidative stress on macular thickness, evaluated by OCT scans, founding a positive correlation with serum level of vitamin C [55]. The attention on the role of oxidative stress in the pathogenesis of DR led the researchers to explore the relation between DR and several molecules known to perform an antioxidant action, such as glutathione, superoxide dismutase, nitric oxide, advanced oxidation protein products [56], L-arginine, L-citrulline [57], and 8-isoprostanes [58]. Although the level of the majority of these markers was significantly higher in diabetes patients compared healthy controls, no differences were found in groups with or without DR.

The serum level of B-vitamins has been measured in five observational studies to investigate their role in T2D retinopathy [59,60,61,62,63]. Nevertheless, also for these vitamins the results are not clear and univocal. A cross-sectional case-control study ruled out by Satyanarayana and colleagues, including 300 T2D patients, demonstrated that plasma level of vitamin B12 was significantly lower in T2D patients with retinopathy compared to those not affected by DR [61]. Moreover, in DR group, higher homocysteine levels were detected. These findings showed a possible association between vitamin B12 deficiency and hyperhomocysteinemia in DR. Instead, no association with DR was observed for vitamin B1, B2, B6, and folate, albeit Horikawa and colleagues recently reported that a high intake of vitamin B6, estimated with a food frequency questionnaire in 978 Japanese T2D patients, was related to a lower 8-year risk of DR occurrence [62], while Cinici and colleagues observed a relation between low level of vitamin B1 and the occurrence of DR [60]. Conflicting data are available regarding the relation between homocysteine level and the course of DR [59,61,63,64]. In two cross-sectional studies, both an increased risk of DR and severity of DR were related to higher homocysteine levels [63], conversely, de Luis and colleagues did not detect statistical differences in the prevalence of DR in T2D patients with normal or high level of homocysteine [64]. Similarly, no univocal results are available o for serum folate [59,61,63].

Melatonin is the main endogenous marker of the circadian rhythm. Since patients with DR could have an impaired retinal light perception and consequent circadian misalignment, some researchers investigated the level and circadian fluctuations in diabetes patients with and without retinopathy [65]. Ba-Ali and colleagues measured salivary melatonin concentration, showing a reduced peak melatonin level and mean nocturnal melatonin concentration both in patients with and without DR, compared to healthy controls, but did not find any significant difference in melatonin concentration between the two groups [66]. Hikichi and colleagues, measuring plasma melatonin levels at midnight and 3 pm, obtained similar results [65]: the night-time melatonin level was impaired in patients with diabetes and PDR but not in those without PDR, suggesting a dysfunction of retinal light perception as a cause of altered melatonin secretion. Nevertheless, these studies did not demonstrate a causal link between melatonin level and the pathogenesis of DR [65,66].

L-carnitine, involved in mitochondrial free fatty acid transport/oxidation, has been reported to improve vascular function [67]. In the case-control study by Poorabbas and colleagues, the level of serum L-carnitine was determined in T2D patients with and without vascular complications, observing a significantly lower level of L-carnitine in patients with DR compared to control group [67].

**Table 1 nutrients-14-04430-t001:** Summary of the selected observational studies focusing on nutraceuticals in diabetes retinopathy.

First Author and Year	Country	Type of Diabetes	Diabetic Groups (*n*)	Control Group (*n*)	Nutraceutical	Sample	Main Results	NOS Scale
Zhao W. J., 2021 [50]	China	T2D	DR (235), NDR (836)	None	25(OH)D	Serum	25(OH)D level was not related to DR	7
Gungor A., 2015 [44]	Turkey	T2D	NPDR (100)	None	25(OH)D	Serum	Reduced mean retinal nerve fibre layer thickness in the group with vitamin D insufficiency	7
Senyigit A., 2019 [68]	Turkey	T2D	DR (30), NDR (133)	HC (75)	25(OH)D3	Serum	Lower 25(OH)D in patients with DM and complications (retinopathy, nephropathy or neuropathy) than the control group and the DM + uncomplicated group	6
Herrmann M., 2015 [45]	Australia New Zealand Finland	T2D	DR (793), NDR (8731)	None	25(OH)D3	Serum	Increased risk of macrovascular and microvascular disease events with low blood 25(OH)D	8
Bajaj S., 2014 [46]	India	T2D	DR (54), NDR (104)	HC (130)	25(OH)D3	Serum	Lower levels of vitamin D in T2D and augmented risk of microvascular complication with vitamin D deficiency	5
Ahmadieh H., 2013 [47]	Lebanon	T2D	DR (32), NDR (104)	HC (74)	25(OH)D3	Serum	Low serum 25(OH)D level was an independent predictor of retinopathy in DM2	4
Kaur H., 2011 [48]	Australia	T1D	DM (517)	None	25(OH)D3	Serum	Increased prevalence of retinopathy in T1D young people with vitamin D deficiency	6
Butler A.E., 2020 [42]	Qatar	T2D	DR (160), NDR (300)	HC (290)	25(OH)D3, 1,25(OH)_2_D3, 24,25(OH)_2_D3, 3-epi-25(OH)D3	Serum	Lower 25(OH)D3 and 1,25(OH)_2_D3 levels related to DR	7
Ahmed L.H.M., 2020 [41]	Qatar	T2D	DR (160), NDR (300)	None	25(OH)D3, 1,25(OH)_2_D3, 24,25(OH)_2_D3, 3-epi-25(OH)D3, VMR1, VMR2, VMR3	Plasma	VMR1 and mainly VMR2 were related to DR	5
Ahmed L.H.M., 2020 [49]	Qatar	T2D	DR (77), NDR (182)	HC (222)	25(OH)D2, 25(OH)D3	Serum	Lower 25(OH)D3 levels were related to DR	5
Millen A.E., 2003 [52]	USA	T2D	DR (199), NDR (799)	None	Vitamin C, Vitamin E	Serum	No significant associations between serum levels of major dietary antioxidants and DR	7
Millen A.E., 2004 [51]	USA	T2D	DR (224), NDR (1129)	None	Vitamin C, Vitamin E, Multisupplements	FFQ	Decreased odds of DR among users of vitamin C or vitamin E supplements or multisupplements	7
She C., 2021 [53]	China	T2D	DR (119), NDR (336)	None	Vitamin C, Vitamin E, Vitamin A, Vitamin B2, Selenium	FFQ	Higher vitamin E and selenium intake appeared to be the protective factors of DR	5
Fahmy R., 2021 [55]	Saudi Arabia	T2D	DR (12), NDR (12)	HC (15)	Vitamin C, TBARS, GSH, GSH S-transferase	Serum	Higher level of vit C and GSH in diabetic vs. controls were predictive of DR	5
Cinici E., 2020 [60]	Turkey	T2D	NPDR (40), PDR (20), NDR (20)	HC (20)	Thiamine Pyrophosphate (Vitamin B1)	Serum	Lower blood TPP concentrations were associated with higher risk of DR	8
Horikawa C., 2020 [62]	Japan	T2D	NDR (978)	None	Vitamin B6	FFQ	High vitamin B6 intake was associated with lower incidence of DR	7
Malaguarnera G., 2015 [63]	Italy	T2D	NPDR (70), PDR (65), NDR (96)	HC (80)	Vitamin B9	Serum	Severity of DR was associated with lower folic acid and red cell folate levels, especially between PDR and NPDR groups	5
Satyanarayana A., 2011 [61]	India	T2D	DR (200), NDR (100)	HC (100)	Folic Acid	Serum	Higher plasma homocysteine levels in T2D patients (especially in the DR group), lower plasma vitamin-B6 and folic acid in the NDR and DR groups than in the HC group	4
Srivastav K., 2016 [59]	India	T2D	NPDR (20), PDR (20), NDR (20)	HC (20)	Vitamin B12, Folic Acid, Homocysteine	Serum	Increased severity of DR and retinal nerve fibre layer thinning were correlated with increased serum levels of homocysteine	6
de Luis D.A., 2005 [64]	Spain	T2D	DR (65), NDR (90)	None	Homocysteine	Serum	Hyperhomocysteinemia in T2D patients was associated with higher levels of fibrinogen, lipoprotein (a), microalbuminuria, and blood pressure	7
Zhou Q., 2018 [54]	China	T1D	DR (34), NDR (103)	HC (50)	Selenium	Serum, Urine	Lower urinary Se levels in T2D patients with DR compared to uncomplicated T2D subjects	6
Yildirim Z., 2007 [56]	Turkey	DM	NPDR (25), PDR (25)	HC (25)	Copper, Zinc, Nitric Oxide, GSH, AOPP, SOD	Serum	Increased AOPP levels in PDR compared to HC	2
Ba-Ali S., 2018 [66]	Denmark	T2D	NPDR (25), NDR (29)	HC (21)	Melatonin	Saliva	Reduced nocturnal melatonin concentration and increased fragmentation of activity-rest intervals in DR group compared to NDR group	5
Hikichi T., 2011 [65]	Japan	T2D	NPDR (16), PDR (14)	HC (26)	Melatonin	Serum	Lower nighttime melatonin levels in PDR group than in the HC and NPDR groups	5
Poorabbas A., 2007 [67]	Iran	T2D	DR (20), NDR (13)	HC (18)	L-Carnitine	Serum	Almost 25% less serum-free L-carnitine levels in DM patients with complications than in those with no complications	5
Hattenbach L.O., 2000 [57]	Germany	DM	DR (14), NDR (8)	HC (20)	L-Arginine, L-Citrulline, N-hydroxy-L-arginine (HOArg)	Aqueous Humour	Higher levels of HOArg in DR and NDR patients than in HC	3
Pękala-Wojciechowska A., 2018 [58]	Poland	T1D	DR (11), NDR (10)	HC (12)	8-Isoprostane	Serum, Exhaled breath condensate	Lower 8-isoprostane in exhaled breath condensate in T1D subjects with advanced complications than in those without advanced complications and in the HC group	6

Abbreviations: NOS, Newcastle-Ottawa; DM, Diabetes Mellitus; T1D, Type 1 Diabetes; T2D, Type 2 Diabetes; DR, Diabetic Retinopathy; PDR, Proliferative Diabetic Retinopathy; NPDR, Non Proliferative Diabetic Retinopathy; NDR, None Diabetic Retinopathy; HC, Healthy Control; GSH, Glutathione; AOPP, Advanced Oxidation Protein Products; SOD, Superoxide Dismutase; TBARS, Thiobarbituric Acid Reactive Substances; DPN, Diabetic Peripheral Neuropathy; DN, Diabetic Nephropathy; VMR, Vitamin D Metabolite Ratio; FFQ, Food Frequency Questionnaire.

**Table 2 nutrients-14-04430-t002:** Summary of the selected interventional studies focusing on nutraceuticals in diabetes retinopathy.

First Author and Year	Country	Type of Diabetes	Diabetic Groups (*n*)	Control Group (*n*)	Nutraceutical	Daily Dose	Intervention Duration	Main Results	NOS Scale
Ellis J.M., 1991 [32]	USA	DM	DM (18)	None	Vitamin B6	50–200 mg	8 months–28 years	No developement or mild form of DR	5
Chatziralli I.P., 2017 [34]	Greece	T2D	NPDR (188), PDR (94)	None	Vitamin E	300 mg	3 months	Serum MDA associated with the severity of DR, Vit. E reduces MDA levels	5
Bursell S.E., 1999 [33]	USA	T1D	NDR (36)	HC (9)	Vitamin E	1800 IU	4 months treatment + 4 months placebo	DM patient retinal blood flow increased and was comparable with that of HC subjects. Normalization of elevated baseline creatinine clearance in DM patients	8
Kheirouri S., 2018 [38]	Iran	DM	DR (50)	None	Zinc	30 mg	3 months	Levels of VEGF, BDNF and NGF were not affected by Zn supplementation	8
Britten-Jones A.C., 2021 [37]	Australia	T1D	NPDR (18), NDR (25)	None	n-3 PUFA (EPA, DHA)	1800 mg (1080 mg, 720 mg)	6 months	In n-3 PUFA group, improvements in central corneal nerve fibre length	8
Sepahi S., 2018 [40]	Iran	T1D, T2D	PDR (60)	None	Crocin (Saffron)	5 or 15 mg	3 months	Crocin 15 mg/die decreased HbA1c and central macular thickness, and improved best corrected visual acuity compared to the placebo group. No significant differences with crocin 5 mg/die.	8
Filippelli M., 2021 [39]	Italy	DM	PDR (28)	None	Curcumin, with or without Homotaurine and Vitamin D3	0.5 and 1 μM, 100 μM, 50 nM	Nothing (in vitro)	Improved TNF-α, IL2, and PDGF-AB with curcumin, homotaurine, and vitamin D3 treatment	6
Domanico D., 2015 [35]	Italy	T2D	NPDR (68)	None	Pycnogenol, Vitamin E, Coenzyme Q10	50 mg, 30 mg, 20 mg	6 months	Reduction in ROS levels and an influence on retinal thickness	6
Parisi V., 2021 [27]	Italy	T1D	NPDR (20)	None	Citicoline 2%, Vitamin B12 0.05% (OMK2)	One drop, thrice daily	36 months	Increased mfERG responses in the treatment group (*n* = 8) with functional enhancement of preganglionic elements located in the 10 central retinal degrees	7
Parravano M., 2020 [28]	Italy	T1D	NPDR (20)	None	Citicoline 2%, Vitamin B12 0.05% (OMK2)	One drop, thrice daily	3 years	A significant reduction in terms of FDT mean sensitivity and in morphology was observed in the placebo group, while no significant changes were observed in the treated group	6
Liu z., 2021 [30]	USA	DM	NPDR (8)	HC (15)	L-methylfolate, Vitamin C, Vitamin D, Vitamin E and others *	1 capsule for the 1st week, 2 capsules for the 2nd week, 3 capsules for the rest of the six months	6 months	In DR MTHFR+, increase in visual acuity and vascular density, the latter particularly in subjects with both A1298C and C677T polymorphisms	6
Liu Z., 2020 [29]	USA	DM	NPDR (8)	HC (15)	L-methylfolate, Vitamin C, Vitamin D, Vitamin E and others *	1 capsule for the 1st week, 2 capsules for the 2nd week, 3 capsules for the rest of the six months	6 months	In DR MTHFR+, axial blood flow velocity, cross-sectional blood flow velocity, flow rate, vessel density were significantly increased compared to baseline	6
Chous A.P., 2015 [36]	USA	T1D, T2D	NPDR (30), NDR (37)	None	Alpha-Lipic Acid, Benfotiamine (Vit. B1), Vit. C, Vit. E and others **	2 capsules	6 months	At 6 months, better visual function, improvements in serum lipids, hsCRP and diabetic peripheral neuropathy in subjects on active supplement compared with placebo	6
Smolek M.K., 2013 [31]	USA	T2D	NPDR (10)	None	L-Methylfolate Calcium, Pyridoxal-5′-Phosphate, Methylcobalamin (Metanx)	2 tablets (3 mg, 35 mg, 2 mg)	6 months	Reduced retinal edema and increased light sensitivity	4

* Vit. B1, Vit. B2, Vit. B6, Vit. B12, Pantothenic Acid, Zinc, Selenium, Copper, N-Acetylcysteine, Lutein, Zeaxanthin (Ocufolin); ** Curcumin, Vit. D3, DHA/EPA, Grape Seed Extract, Green Tea Extract, Resveratrol, N-Acetyl Cysteine, Coenzyme Q10, Zinc, Pycnogenol, Lutein, Zeaxanthin. Abbreviations: NOS, Newcastle-Ottawa; DM, Diabetes Mellitus; T1D, Type 1 Diabetes; T2D, Type 2 Diabetes; DR, Diabetic Retinopathy; PDR, Proliferative Diabetic Retinopathy; NPDR, Non Proliferative Diabetic Retinopathy; NDR, None Diabetic Retinopathy; HC, Healthy Control; MDA, Malondialdehyde; hsCRP, High-sensitivity C-reactive protein; ROS, Reactive Oxygen Species; VEGF, Vascular Endothelial Growth Factor; BDNF, Brain Derived Neurotrophic Factor; NGF, Nerve Growth Factor; mfERG, Multifocal Electroretinogram; FDT, Frequency Doubling Technology; MTHFR, Methylenetetrahydrofolate Reductase.

## 4. Discussion

DR is a fearsome complication of DM, which can lead to vision loss and impaired quality of life. DR progresses from mild, subclinical abnormalities to moderate or severe stages characterised by retinal ischaemia, new vessels development and relevant visual impairment or loss [69]. The current treatment of DR is based on invasive and expensive procedures (laser photocoagulation, vitrectomy, intravitreal drugs) for the management of the advanced stages of the disease, while few strategies are available for the early stage or prevention, when the retinal loss is minimal or absent and a therapeutic intervention could be most effective in stopping the progression of retinal damage, avoiding a sight-threatening retinopathy [69]. In this scenario, with the pandemic rise of DM and of individuals at risk of DR-induced vision loss, new non-invasive and less expensive therapeutic strategies, associated with screening programs, would be desirable. Despite chronic hyperglycaemia and long-term glucose control being the main determinant of this highly specific vascular injury related to DM, other extraocular factors, such as hypertension, dyslipidaemia, obesity, cigarette smoke, pregnancy, genetic and epigenetic susceptibility, contributes to the onset and course of DR [11]. Nevertheless, some diabetes patients without these traditional risk factors can likewise develop a retinal damage, suggesting the involvement of other pathogenetic elements [70]. Preclinical and clinical studies investigated whether nutraceuticals could influence the course of DR [23,24]. Moreover, diabetes itself can alter the nutritional status, suggesting an attitude to deficiency of micronutrients in diabetes patients [61].

Our systematic review of clinical studies summarizes the emerging evidence on the role of nutraceuticals in DR. The most promising results concern vitamin B, D, and E.

Vitamins B have been widely studied in both observational and intervention studies (Table 1 and Table 2). Only few investigations observed the effect of single vitamin B on various retinal outcomes, while most of the studies, evaluated the complex of vitamins B. Although this approach, compared to the administration of single vitamins, allows for the enhancement of the potential beneficial effect of multiple vitamins, it makes difficult to detect each vitamin size effect. In observational studies, lower plasma levels of vitamin B1 and B12 were related to a higher occurrence of DR [60,61]. In interventional studies, the treatment with vitamin B12 determined improvements of the macular bioelectrical responses [27] and stabilization of functional impairment, neuroretinal degeneration, and microvascular damage [28]. In particular, the ocular application of vitamin B12 and citicoline in T1D patients demonstrated a protective role on plexiform layer thickness, foveal vessel density, and macular bioelectrical responses detected by multifocal electroretinogram. It is well known that the oxygen rate consumption of photoreceptors is very higher compared to other cells of the organism, thus these cells are exposed to a high degree of oxidative stress [24]. Hence, the beneficial effect of vitamin B12 in DR could be mediated by ROS reduction and release of neurotrophic and neuroprotective growth factors [24]. Moreover, vitamin B12 improves, in rats, retinal hypoxia, VEGF overexpression, and reduces endoplasmic reticulum stress-mediated cell death [71]. Vitamins B complex, alone or in combination with other vitamins (C, D, E, etc.), improved visual acuity, light sensitivity, retinal oedema and vessel density [29,30,31] (Table 2).

Oxidative stress reduction is a cornerstone in the approach to DR. The antioxidant properties of vitamin E could play a role in the course of DR, as suggested by the reduction in MDA and ROS level in T2D patients with DR treated with at least for three months with vitamin E [34,35]. In addition, high-dose vitamin E (1800 IU vitamin E) has demonstrated the ability to improve both retinal blood flow, mainly in patients with lowest retinal blood flows at baseline and poorest glucose control, and central macular thickness [33,35]. The effect on retinal blood flow was due to the vitamin E storage in lipid-rich retinal compartments and following normalization of the diacylglycerol/protein kinase C signalling trough the activation of DAG diacylglycerol kinase [33].

Vitamin D is the most investigated vitamin in observational studies included in our systematic review (Table 1). The existing data highlighted a higher risk of DR in diabetes patients with lower level of vitamin D [41,42,43,44,45,46,47,48]. Potential mechanisms underlying the relationship between vitamin D deficiency and DR course include β-cell dysfunction—possibly due to reduced intracellular calcium concentration—insulin resistance, chronic inflammation, and endothelial dysfunction [45]. Nevertheless, the results are not always univocal and, importantly, are not confirmed by intervention studies, which investigated vitamin D in complex, making it difficult to establish the real contribution of every single element.

Initial evidence available concerning the effect of antioxidant natural substances (e.g., curcumin, homotaurine, crocin) on the pathogenesis of DR. Other RCTs, with a larger sample, are necessary to clarify this item.

## 5. Conclusions

The multifactorial pathogenesis of DR, involving the interplay between the detrimental effects of hyperglycaemia, dyslipidaemia, hypoxia, and ROS, provides several pathways potentially targeted by nutraceuticals. The current evidence is not robust enough to routinely propose nutraceutical in the clinical approach to DR. Moreover, the heterogeneity of the included studies did not allow us to investigate the role of specific nutrients on DR. For this reason, further investigations, as well as specific systematic reviews, are required to summarize and confirm the efficacy of nutrients and nutraceuticals for the prevention and treatment of DR. First, most of the existing studies were carried out involving diabetes patients with retinopathy already diagnosed. Despite this, little evidence is available on the efficacy of nutraceuticals at different retinopathy stages: are nutraceuticals also effective in the advanced stages of DR or is their application is limited to the early stages or in prevention, even in selected subjects particularly at risk, when DR is absent? Second, further RCTs are necessary to establish the most effective combination of nutraceuticals, which offers a synergistic protection against several pathogenic mechanisms contributing to DR, the dosage and duration of the supplementation to achieve clinically meaningful benefits. Lastly, most of the evidence derived from cohort of T2D patients, while very few data are available on their use in T1D, whose pathogenesis differs from T2D. In conclusion, further research is needed to better investigate the role of nutrients on DR risk and progression, and to evaluate if nutraceuticals supplementation could be considered as a valid approach for the management of DR. The hope is that emerging therapeutic strategies, in addition to conventional ones, and the last but not the least, a broad and appropriate use of screening procedures for DR, could be effective to counteract the diabetes pandemic and thereby decreasing the number of individuals with retinal damage and vision loss.

## Figures and Tables

**Figure 1 nutrients-14-04430-f001:**
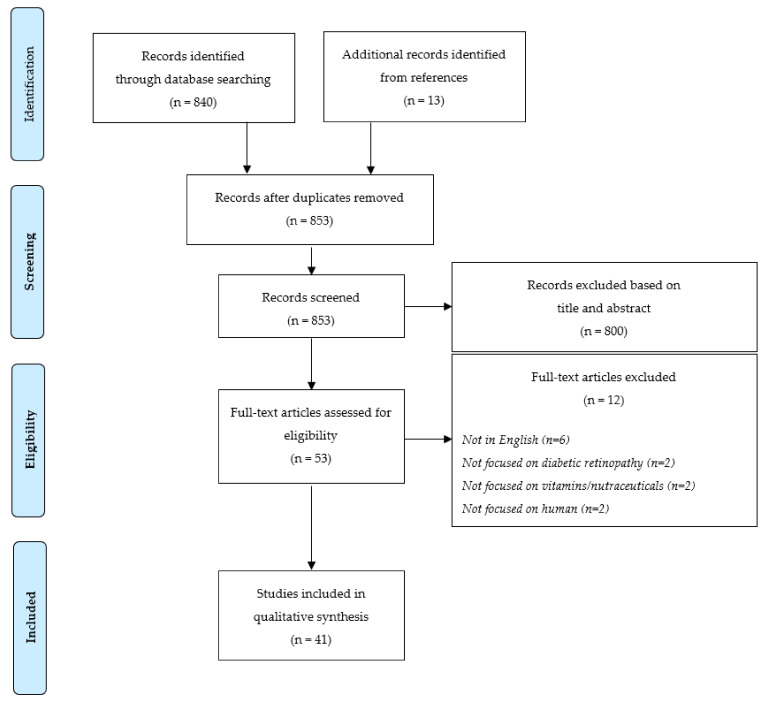
PRISMA 2009 Flow Diagram of study selection.

## Data Availability

Data used to support the findings of this study are available from the corresponding author upon request.

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
