# Peer review of "Do Nutrients and Nutraceuticals Play a Role in Diabetic Retinopathy? A Systematic Review"

_nutrients, 2022, doi:10.3390/nu14204430_

Round 1
Reviewer 1 Report
Dear authors, I read manuscript named “Do Nutraceuticals Play a Role in Diabetic Retinopathy? A Systematic Review” with great interest. My basic impression is that you have lost focus in trying to present a lot of things related to nutrients and the occurrence of diabetic retinopathy. It was difficult for me to keep track of those numerous nutrients (whether their levels in biological samples or effects from dietary intake or supplementation were evaluated) and then relate results to types of diabetes, various stages of retinopathy, and outcomes of included studies.
Here are some specific comments:
Line 55. “In this systematic review of clinical studies, we provide an updated summary of studies investigating the impact of vitamins and nutrients in DR.”
This manuscript provide summary of studies without focused clinical question so the type of the manuscript can be only narrative review.
Line 57. “Materials and Methods”
Outcome measures were not predefined.
Line 139.
Assessment of the quality of the included studies and the overall strength of the evidence should be given.
Line 160. “The effect of Ocufolin, a vitamin complex composed by L-methylfolate (vitamin B9), vitamin B1, B2, B6, B12, C, D, E, N-acetylcysteine and other compounds (Table 2), on DR was tested in a specific cohort of diabetes subjects with methylenetetrahydrofolate reductase (MTHFR) polymorphisms [29,30]..... Liu and colleagues, in a small prospective cohort, observed that after six months of Ocufolin administration, patients with MTHFR polymorphisms..”
Written like this, Liu and colleagues study looks like another study.
Line 162. “diabetes subjects”
Whether they were patients with T1D or T2D?
Line 166. “prospective cohort”
Did you mean prospective interventional cohort study?
Line 232. “in most of these studies “
I suggest writing exactly how many out of 10.
Line 250. “In particular, the authors calculated the following ratios: 250 24,25(OH)2D3:25(OH)D3, 1,25(OH)2D3:25(OH)D3, 3-epi-25(OH)D3:25(OH)D3. DR was 251 significantly associated with both 24,25(OH)2D3:25(OH)D3 and 1,25(OH)2D3:25(OH)D3 252 ratio.”
Unclear
Line 255.
How many observational studies investigated the influence of vitamin C I E on the onset or progression of DR, and what was found?
Line 258. “serum vitamin E and C”
Did you mean serum vitamin E and C levels?
Line 258. “higher intake of both vitamin”
Did you mean from food?
Line 276. “in many observational studies”
I suggest writing exactly how many.
Line 276. “the level”
Did you mean serum melatonin levels?
Line 333. “human studies”
It should be replaced with a more appropriate term: clinical studies
Line 341. “The positive effect of vitamin B12 on DR observed in observational studies”
This statement is not correct because in presented observational studies the serum level vitamin B12 has been measured not effect.
Line 392: “In conclusion, nutraceuticals supplementation could be a promising approach for the management of DR.”
A conclusion like this cannot be made based on results of most observational studies which evaluated the amount of vitamins/minerals in biological samples or their dietary intake.
Line 393: “The hope is that emerging therapeutic strategies, in addition to conventional ones, and the last but not the least, a broad and appropriate use of screening procedures for DR, could be effective to counteract the diabetes pandemic and thereby decreasing the number of individuals with retinal damage and vision loss.”
I suggest deleting this part of the conclusion because it is not related to the topic.
Line 413: “12. Retinopathy...”
Delete number 12
Reviewer 2 Report
Comment to authors:
Thanks for the opportunity to review the manuscript entitled ‘Do Nutraceuticals Play a Role in Diabetic Retinopathy? A Systematic Review’. This study was a systematic review of clinical studies investigating the impact of vitamins and nutrients on DR, including 41 studies. The systematic review indicated that nutraceuticals supplementation could be a promising approach for the management of DR. Specific concerns are as follows:
1. Study type: With 41 studies included in the review, 27 were observational and 14 - eight of them randomized and controlled–intervention included in the analysis, did the authors try to carry out a meta-analysis? Which could provide a more quantitative and straight outcome.
2. Line 61, Methods-literature search: The database searches were incomplete. Did the authors searched EMBASE? For example, a meta-analysis of observational studies published in 2017, fifteen studies regarding the association between vitamin D deficiency and diabetic retinopathy in type 2 diabetes were included (PMID: 28335514).
3. Line 66-69, the Search Strategy was ambiguous, which might lead to the missing of some relevant articles. Terms like (retinopath* or ?edema* or macula* or maculopath* or microaneurysm* or neovascular*) should also be considered.
4. Line 77, quality assessment: heterogeneity between studies and potential publication bias were not clarified. Risk of bias graphs should be included in the study.
5. Line 86, confoundings: age, gender and diabetes duration are important factors related with DR. The basic characteristics of the mentioned information should be presented. How about the racial difference?
6. Line 142, results: except just presenting the outcomes from included studies, comparison between different studies, especially those with same nutraceuticals should be discussed. Furthermore, conclusive and precise findings should be provided.
